# Genetic Modifiers Associated with Vaso-Occlusive Crises and Acute Pain Phenomena in Sickle Cell Disease: A Scoping Review

**DOI:** 10.3390/ijms26094456

**Published:** 2025-05-07

**Authors:** Froso Sophocleous, Natasha M. Archer, Carsten W. Lederer

**Affiliations:** 1Molecular Genetics Thalassaemia Department, The Cyprus Institute of Neurology & Genetics, 6 Iroon Avenue, Ayios Dometios, Nicosia 2371, Cyprus; sfroso@hotmail.com; 2Pediatric Hematology/Oncology, Dana-Farber/Boston Children’s Cancer and Blood Disorders Center, Harvard Medical School, Boston, MA 02115, USA; natasha.archer@childrens.harvard.edu

**Keywords:** GWAS, sickle cell disease, vaso-occlusive crises, acute chest syndrome, priapism, dactylitis, genetic disease modifier, hemoglobinopathy, avascular necrosis, splenic sequestration

## Abstract

Sickle cell disease (SCD) is a group of recessive diseases caused by the β^S^ sickling mutation of *HBB* in homozygosity or in compound heterozygosity with other pathogenic *HBB* mutations. Patients with severe SCD typically experience painful vaso-occlusive crises and other pain-related phenomena, including acute chest syndrome, priapism, dactylitis, avascular necrosis, and splenic sequestration and infarction. High variability of pain-related phenomena per SCD genotype indicates genetic disease modifiers (GDMs) as pathology determinants and, thus, as critical to prognosis, treatment choice, and therapy development. Articles likely holding genetic information for SCD pain phenomena were identified in PubMed and SCOPUS for article quality assessment and extraction of corresponding GDMs and observations indicative of development areas in our understanding of SCD GDMs. This process led to the initial selection of 183 articles matching the search terms, which, after two-step selection, resulted in the inclusion of 100 articles for content analysis and of significant findings for GDMs from 37 articles. Published data point to gender effects and to 51 GDM SNVs, deletions, and regions, including globin genes and significant overrepresentation of gene ontology pathways related, e.g., to oxidative stress, hypoxia, and regulation of blood pressure. Analyzed articles further pointed to additional candidate GDMs affecting SCD VOC and pain phenomena and to potential confounding factors for GWAS analyses. We found that despite the critical importance of VOC and pain phenomena for SCD pathology, corresponding clinically relevant genetic insights are held back by a shortage of large-scale, systematic multi-ethnic efforts, as undertaken by the INHERENT Network.

## 1. Introduction

Sickle cell disease (SCD) is characterized by vaso-occlusion and chronic hemolysis [1], brought about by a qualitative defect in erythroid β-globin (HBB) expression. The sickling β^S^ HBB^E7V^ variant results in abnormal, sickling hemoglobin heterotetramers (α_2_β^S^_2_, HbS), which take the place of normal adult hemoglobin (α_2_β_2_, HbA) in erythroid cells, where they precipitate under low oxygen pressure, thus leading to a change in cell shape and occlusion of capillary vessels by the resulting sickle-shaped cells. Corresponding ischemia and consequential vaso-occlusive crises (VOC) represent the primary source of morbidity in SCD and are the most frequent cause of hospitalization for SCD patients [2]. Sites of VOC and pain phenomena vary greatly by age, between but also within subjects over time, and be it for pain in the chest, back, limbs, joints, or the head as frequent sites, understanding the causes is instrumental to effective SCD management [3,4]. Ischemia brings about inflammation and necrosis of different tissues, including osteonecrosis and avascular necrosis (AVN), so that over time, recurrent pain episodes may develop into chronic pain, e.g., through persistent tissue and neuronal damage. Moreover, VOC, ensuing necrosis, bone marrow infarcts, and resulting fat emboli can cause acute chest syndrome (ACS), which may quickly lead to intubation and represent a frequent cause of death in SCD patients [5]. Sickling may also block the blood supply to capillaries in hands and feet, typical of early manifestations of SCD in children, and lead to dactylitis, with painful local swelling and inflammation. Similarly, sickling may block penile blood supply, leading to priapism and ischemic tissue damage and pain, impairing sexual function. It may also lead to insufficient oxygen supply to the bones, leading to AVN, which in the leg typically affects the head of the femur and, in the arm, the humerus. A potentially life-threatening and extremely painful consequence of sickle cell blockage of blood flow is splenic sequestration, when trapped sickled cells in the spleen may quickly lead to splenic enlargement, abdominal pain, and even death due to limited circulating blood cells. Severe episodes require splenectomy to avoid recurrent sequestration. In the absence of splenic sequestration, atrophy of the spleen secondary to splenic infarction is near universal and leads to an increased risk of infection. Finally, sickle cells may also impair the blood supply to the retina, causing ischemia, pain, and eventually, retinal damage and impaired vision. All of these phenomena are interrelated, linked by blockage of blood vessels and pain in SCD, and may, therefore, be ameliorated or aggravated by a shared subset of genetic disease modifiers (GDMs).

Several GDMs have been long-established for SCD, which, in the case of γ-globin and α-globin as hemoglobin components, was the result of targeted mutation detection and noting of corresponding disease phenotypes. Of these SCD GDMs, γ-globin was early-on recognized as having anti-sickling properties, based on substitution of β^S^ in hemoglobin to form fetal hemoglobin (α_2_γ_2_) and on molecular interactions of its alternative amino acid residues with existing HbS heterotetramers, effectively reducing stacking of hemoglobin molecules and preventing cell sickling [6]. Hypothesis-driven investigations likewise led to the identification of nitric oxide (NO)-related enzymes as important modifiers of SCD severity, as hemolysis in SCD releases free heme, which scavenges NO and thus prevents its role in vasodilation, anti-thrombotic, anti-adhesion, and anti-oxidant action, all of which directly relate to SCD complications [7]. In the same vein, vascular endothelial growth factor (VEGF) has a key role in angiogenesis and is upregulated in SCD, where it functions to reduce ischemia while also favoring oxidative stress and inflammatory responses [8]. The role of many other GDMs, however, cannot readily be concluded from their known roles and pathophysiological mechanisms. For instance, *BCL11A* encodes a transcription factor and master switch for β-globin activation and γ-globin inactivation but was initially characterized as a chromosomal translocation site for B-cell non-Hodgkin lymphoma [9]. It’s SCD GDM role and potential utility as a therapeutic target for SCD was only discovered by genome-wide association studies (GWAS), linking *BCL11A* mutations to elevated γ-globin levels and, thus, by inference to milder SCD-related VOC and pain phenotypes [10]. It can be expected that suitably designed GWAS and other hypothesis-free approaches will allow the discovery of many more GDMs of VOC and pain phenomena, and this scoping review sets out to chart the known knowns and the known unknowns as a waypoint to advanced study designs and further discoveries [11].

## 2. Materials and Methods

This scoping review follows the PRISMA Extension for Scoping Reviews and additional practice guidelines [12,13], modified to allow the inclusion of additional articles from secondary searches. Articles of interest were identified by searching the following search terms in the PUBMED (https://pubmed.ncbi.nlm.nih.gov) and SCOPUS (https://www.scopus.com/home.uri) online databases: pain* AND (vaso-occlus* OR “acute chest syndrome” OR priapism OR dactylitis OR osteonecrosis OR “avascular necrosis” OR splenic OR retinal OR retinopathy) AND sickle AND (genome-wide OR SNV OR “genetic modifi*” OR “risk score” OR “causative mutation” OR polymorphism OR variation OR variant). The last updated search was performed on 1 May 2024. The Rayyan AI online tool (https://rayyan.ai/reviews) was used to identify duplicates, as well as the different types of articles included in our exports from the two databases. Then, manual curation of the article list for actual analysis of phenotypic terms and a detailed quality assessment of shortlisted articles were performed. After initial discussion of uncertainties in the interpretation of Joanna Briggs Institute (JBI) critical appraisal tools (https://jbi.global/critical-appraisal-tools) for the literature in hand, parallel assessment by two independent reviewers of 20 of the primarily identified original studies for calibration showed a high level of concordance (with 3 articles evaluated as “Include” vs. “Seek further info” as apparent discrepancies), so that the assessment of additional articles was divided between both reviewers. In the process, the JBI critical appraisal tools were used for cohort studies, case reports, case-control studies, and systematic reviews and research syntheses. Articles found of sound methodology and addressing relevant phenotypes were labeled *Include* and included in the text, of which those with significant findings for GDMs and relevant phenotypes were also included in Table 1 on SCD pain phenotype-associated GDMs. Of the articles labeled *Exclude*, some were still included in text sections for which they were informative, as indicated in the Comments of the corresponding appraisal forms (see Appendix A). Formulation of summary and synthesis findings were performed by CWL and FS and revised by NMA. Methodology for identification and inclusion/exclusion is summarized in Figure 1.

For easier accessibility of the text, variations in reporting between source articles were harmonized for this review. In this vein, consensus terms were used for varying expressions referring to the same or similar concepts, including vaso-occlusive crisis (VOC) also for the expressions vaso-occlusive episode (VOE), vaso-occlusive pain (VOP) and (acute care) utilization, and avascular necrosis (AVN) also for the expression osteonecrosis (ON). Likewise, incomplete reporting of *p*-values in source articles was harmonized so that references to *p* < 0.000 are reported here as *p* < 0.0001. Finally, harmonization of age parameters across studies was not possible without losing information, so that where available, age ranges or standard deviations of the mean for ages of patient and control populations are reported here as originally published.

We also considered reviews identified by our original search to retrieve relevant original research articles that filtration by search terms may have missed, in order to avoid inadvertent omission of informative studies. Such secondarily identified articles (including original studies and systematic reviews) were identified by searching originally retrieved reviews for keywords, keyword fragments, and acronyms, and by inspecting original articles referenced in the keyword context in conjunction with reference to specific genes, variants, proteins, or protein complexes. Data for other types of reviews or for poster-only references were excluded. Articles not linked to specific genetic variants but establishing clear links between genes or proteins and pain-associated phenomena are indicative of areas underexplored for genetic variants and are summarized in the corresponding Section 3.11. Case studies contributing information for globin genotypes and for environmental factors were also included in the corresponding Section 3.2 and Section 3.10, respectively. By contrast, articles linked to specific non-globin variants underwent quality assessment as for originally identified primary articles to allow inclusion in the present article. For systematic reviews listing specific variants, summary data for genetic variants and pain-associated phenomena were extracted without formal evaluation of source articles by inclusion in Appendix A. Finally, the full list of genes corresponding to the identified genetic variants was used as input for the STRING database v12.5 (www.string-db.org, accessed 14 April 2025) [15].

## 3. Results and Discussion

### 3.1. Overview of Articles Included in This Review

The selection procedure of articles for inclusion (see Figure 1) led to a full-text inspection of 100 articles. Of these, 52 passed inspection for quality and relevance, with identification of significant findings from 37 articles for inclusion, representing gender effects and 51 GDM deletions, SNVs, haplotypes, or regions, for inclusion in Table 1. In addition to such articles with statistically significant GWAS findings, additional articles were included in the main text of this article based on their relevance for the interpretation of potential GDMs or for the future design of corresponding GWAS studies. Strikingly, some cohorts and studies contributed disproportionately to the number of entries in Table 1, such as the US Cooperative Study of Sickle Cell Disease (CSSCD) cohort, which alone contributed data to 5 entries, while all countries from Africa, though highest in absolute and relative SCD disease burden, together contributed only 4 studies, likely reflecting the resource dependence of GWAS studies and a need for greater inclusiveness in the identification of GDMs.

### 3.2. Globins as Overriding Determinants of SCD Severity

#### 3.2.1. The Influence of γ-Globin and α-Globin

Variants affecting α- and γ-globin levels or function have long been understood as fundamental genetic modifiers of disease pathology, though many studies investigating either do not specifically refer to VOC or pain phenomena in their statistical analyses. Suffice it to say that any variants elevating γ-globin levels alleviate disease severity for both β-hemoglobinopathies, so that the wealth of studies and findings on genetics [16,17,18] target genes for therapy development [19,20,21] and ongoing efforts to identify new genetic modifiers, such as INHERENT [22], indirectly also indicate genetic factors that will reduce pain in SCD. For α-globin and factors modulating its expression, the picture is more nuanced than for γ-globin [23]. Whereas in β-thalassemia, the ratio of β-like-to-α-globin chains is critical so that co-inheritance of α-thalassemia causative variants consistently leads to milder disease phenotypes for a given β-globin locus genotype [24], in SCD, carrier status for α-thalassemia has a mixed effect on different aspects of SCD pathology, such as VOCs, vasculopathy, and blood viscosity [25,26]. As early as 1982, a study on SCA patients with normal (αα/αα), α–/αα, and α–/α–*HBA1*/*HBA2* genotypes investigated the interaction of α-thalassemia with SCA, where odds ratio (OR) calculation of the numbers clearly indicate that absence of two α-globin genes had a beneficial effect for the reduction of ACS (*p* = 0.0161, with events:non-events for αα/αα SCA 30:58 and for α–/α– SCA 7:37) [27]. Likewise, in a study based on the long-established US CSSCD cohort [28] and focused on the identification of priapism-associated single-nucleotide variants (SNVs) in other genes, the subgroup of SCA patients carrying α-thalassemia alleles were less likely to have priapism (*p* = 0.05) than those with αα/αα genotype [29]. In a French study with 314 children with SCA, single-gene–deletion alleles for α-globin (specifically, the -α3.7k deletion) were significantly enriched in the VOC group (*p* < 0.001), in agreement with findings for -α3.7k in an SCA cohort (*n* = 436) in Cameroon [30], whereas they were depleted in the vasculopathy group (*p* < 0.001) [26]. An earlier Cameroonian study investigated the correlation between α-thalassemia, hematological indices, and clinical events in SCA patients (*n* = 161; 17.5 (11–24) years) and *n* = 103 controls (HbAS (24 (17.5–26) years) and HbAA (26.5 (23.2–30) years)) [31] and showed that co-inheritance of α-thalassemia and SCA was associated with improved hematological indices, i.e., increased median RBC count (*p* = 0.01) and decreased median MCV and WBC counts (*p* < 0.0001 for both), as well as lower consultation rates (*p* = 0.038) [31]. Other small-scale studies instead showed no impact of α-globin genotype on pain crises (*n* = 75, of which 31 had α-thalassemia trait) [32] or variably showed an increased frequency of adverse events, such as VOCs [30,33,34] and AVN [35], or reduced frequency of ACS [34], in the presence of α-thalassemia mutations. Surprisingly, given the high frequency of co-inheritance of α-thalassemia variants in SCD, no large-scale systematic study has been dedicated to investigating this complexity and analyzing specific *HBA1*/*HBA2* variants or the overall effect of *HBA1*/*HBA2* variants on SCD pathophysiology and pain.

#### 3.2.2. The Influence of HBB Alleles

SCD spans a range of genetic conditions, with subgroups including sickle cell anemia (HbSS), hemoglobin Sβ^0^, hemoglobin Sβ^+^, hemoglobin SC disease (HbSC), and hemoglobin SE disease (HbSE) as common genotypes, and combinations of other, rarer *HBB* variants with the β^S^ allele. The influence of those combinations of *HBB* alleles has a profound effect on the severity of the disease phenotype and SCD-related pain and, therefore, on the effect of any related GDMs. For instance, in a multi-center study designed to investigate the clinical and genetic associations with priapism within a large Brazilian cohort of *n* = 1314 male patients, of which *n* = 188 (27% < 18 years) had the condition, priapism was more prevalent among more severe SCD genotypes (*p* < 0.0001) such as HbSS, in addition to correlating with age (*p* = 0.006) [36].

One factor in determining disease severity is the level of β-globin reduction in HbS/β-thalassemia compound heterozygotes, which may, at times, obscure symptoms and pain phenotypes enough to greatly delay a clear molecular diagnosis [37]. Another factor is the concomitant activation of γ-globin on the β-globin locus and corresponding HbF levels. For combination with β-thalassemia, HbF levels were generally elevated for different HbSβ^+^ (5.3–15.3%) and HbSβ^0^ (13.1–35.3%) genotypes [32], though highly variable depending on the β-thalassemia mutation. For other combinations, HbF levels ranged from 21.8% (Hb 14.7 g/dL) to 34% (Hb 11.7 g/dL) in rare HbS/Black(Aγδβ)^0^-thalassemia (*n* = 9) [38], from 16.6% to 34.7% for HbSD-Punjab, from 0.6% to 20.3% (median 6.7%, *n* = 13) for HbSO_Arab_ [39], and were slightly elevated in HbSβ(Quebec-Chori) (HbF 9.3%) [40]. High HbF levels generally correlate with milder SCD and pain phenotypes [32], contributing to the similarity of HbSO_Arab_ with HbSS in VOCs, ACS, and dactylitis [39]. However, HbSβ^0^ patients, despite high HbF levels, may nevertheless present with severe pain crises [32].

In addition to the level of β-globin reduction and γ-globin induction, hemoglobin variant proteins also exert effects in patients with SCD by aggravation of, non-contribution to, or amelioration of sickling. In this context, a proportion of HbS ≤ 75% of total hemoglobin already serves to reduce the risk and frequency of VOC events [41]. Another key aspect is the quality and quantity of the variant protein for HbSS and compound-heterozygous HbS genotypes. For example, pediatric HbSC patients have only moderate elevation of HbF levels (of 4.9% (*n* = 461) [42] and of 2.1% (*n* = 96) in two different studies), lower than HbSS (6.2%) patients (*p* < 0.001) [43], but they still register only half as many pain episodes as HbSS patients [42,44] and much lower splenic sequestration (*p* = 0.0001) and ACS occurrence (*p* = 0.005) [44], which is attributed to diminished contribution of HbC to sickling. In the same vein, the amino acid composition of the δ/β fusion product in Hb Lepore leads to an anti-sickling effect and, thus, to milder VOC phenotypes in HbS/β(Lepore) patients [45]. On the other end of the spectrum, high disease severity is brought about by the stabilization of sickling with HbSS and HbSO_Arab_ genotypes [39]. However, across β-globin genotypes and where numbers of affected individuals allow the comparison, it is clear that the frequency and severity of pain-related SCD symptoms are, in general, highly variable between patients with the same genotype and even between siblings [38,46,47], just as HbF levels are highly variable between reports for different HbSS populations, from 6.2% in a Brazilian cohort of African descent [43] to 24.4% in an Indian SCA cohort [48], pointing to factors beyond *HBB* and *HBG1/2* that affect severity and pain in SCD.

Several studies also considered β-globin haplotypes as a parameter influencing HbF expression and, with it, SCD severity [33,41,44,46,49,50,51], but it appears that the β-globin haplotype as a shorthand for conflating six SNVs on the β-globin locus is a suboptimal means of resolving the influence of modifiers on the disease parameters under study here. Accordingly, many of these studies do not establish a significant association of β-globin haplotypes with disease phenotypes. In one such study, patients with HbSD-Punjab and the same (Bantu) haplotype (*n* = 12) showed greatly variable disease severity, from asymptomatic to severe, and appearance of pain phenomena, with painful crises, ACS, priapism, and AVN occurring in 8, 2, 1, and 1 patients, respectively [46]. In a Brazilian pediatric cohort, a mixed population of 71 HbSS and HbSC patients showed no difference in pain events or ACS according to haplotype [44]. In another study (*n* = 190) based on HU and placebo treatment, no effect was detectable for β-globin locus haplotypes in the HU treatment group (*n* = 94) [33]. However, for the corresponding placebo group (*n* = 96), what are considered favorable haplotypes (SEN/SEN, BEN/SEN, CAM/SEN, or SEN/ATYP), remarkably, were associated with increased pain events (*p* = 0.02) and dactylitis events (0.0002) compared to unfavorable haplotypes (CAR/CAR, CAM/CAR, CAR/ATYP, or BEN/CAR). By contrast, in the same group, the α-thalassemia trait and *BCL11A* SNVs showed reduced pain and dactylitis events [33], emphasizing the pronounced effect of factors altogether outside the β-globin locus.

### 3.3. Genetic Modifiers Associated with Pain in Vaso-Occlusive Crisis as the Main Phenotype

Numerous studies showed a clear correlation of variants with VOC phenotypes in patient cohorts. For example, in a pediatric Brazilian SCA cohort (*n* = 117; 5 ± 3 years), an association was shown between *MBL2* polymorphism, low MBL serum production, and frequency of VOCs (*p* = 0.0229) [52]. With a role in innate immunity recognition, *MBL2* was also investigated for both respiratory tract infections (RTIs) and VOCs in a uniformly vaccinated Brazilian pediatric cohort (*n* = 89; 2.5 (0–5) years) to show that deviation from the normal sequence for the A/O haplotype (rs5030737 (R52C), rs1800450 (G54N), rs1800451 (G57E)) in homozygosity or heterozygosity resulted in enrichment for the occurrence of VOCs (*p* = 0.039), but with no measurable effect on RTIs [53]. In a follow-up study for the same cohort one year later, a combined haplotype of said A/O haplotype with the X/Y promoter genotype (referring to an *MBL2*-221G > C SNV), where the YA/YA genotype is associated with high MBL2 expression, revealed lower age-normalized VOC frequency for YA/YA patients (*p* = 0.0188) [54]. The role of MBL2 in the VOC pathophysiology should thus be explored through further studies.

A whole-genome sequencing study of *n* = 722 individuals with HbSS or HbSβ^0^, derived from two different sources, built a polygenic score for VOC in pediatric (>11.5 years) SCD [55]. An association was shown between VOC and 21 SNVs from 9 different loci, including genes that regulate erythrocyte HbF (*BCL11A*, *MYB*, and the β-like globin gene cluster) and genes that were previously linked to pain syndromes (*COMT*, *TBC1D1*, *KCNJ6*, *FAAH*, *NR3C1*, and *IL1A*) [55]. An unweighted polygenic score for all the VOC-associated SNVs showed a strong association with VOC event rate and occurrence (*p* < 0.0001), giving insight into VOC genetic modulation in children with SCD [55]. In another SCD study (*n* = 136), an elevated VOC rate was specifically shown for three minor *NR3C1* SNP alleles (rs33389, rs2963155, rs9324918) [56].

The modifying effect of HbF-associated genetic polymorphisms (i.e., in the γ-globin promoter, *BCL11A*, and *HMIP*) in *n* = 417 SCD patients > 2 years old who received hydroxyurea (HU, aka hydroxycarbamide) treatment (last treatment age: 12.6 ± 8.5), was studied through a German registry [57]. There was no SNV-to-VOC association; however, HU-treated patients with γ-globin promoter polymorphism demonstrated higher frequencies of painful crises and hospitalizations when compared to patients without this polymorphism (*p* < 0.01) [57].

The relationship between *COMT* SNVs with patient self-reported pain, number of acute VOCs, and daily life impact and health care utilization were assessed in HbSS African American patients (*n* = 438; 36 ± 12.5 years) as part of the walk-PHaSST study [58]. An association between a *COMT* haplotype with the risk alleles rs4633 and rs165599 (giving an ATCAA haplotype sequence) with increased pain-related ER visit frequency was indicated (*p* = 0.0004), with this observation being more profound in women (*p* = 0.0001) [58]. Therefore, rs4633 and rs165599 *COMT* variants may predispose women with SCD to worse acute pain.

A GWAS study was conducted to identify acute severe VOC pain variants in African pediatric SCA patients from the CSSCD (*n* = 359, age at enrolment: 2–12 years, with follow-up of 2.8 ± 0.67) and the SIT (*n* = 934; age at enrolment: 2–15 years, with follow-up of 3.0 ± 0.0 years) cohorts [59]. A novel locus was identified to have genome-wide significance (*p* < 0.0001), i.e., SNV rs3115229 located upstream of the *KIAA1109* gene on chromosome 4 [59]. This locus included the *KIAA1109*-*TENR*-*IL2*-*IL21* linkage disequilibrium block, which had previously been associated with autoinflammatory diseases (e.g., celiac disease, ulcerative colitis, and rheumatoid arthritis), indicating that further mechanism-based studies are required.

The following study provided evidence that could support the development of a genetic risk model for painful VOC in Cameroonian SCD patients. A diseased cohort of *n* = 436 HU- and opioid-naïve patients (16 (5–54) years) and a control cohort of 105 matched individuals (HbAS and HbAA) were included [30]. Female sex, body mass index, Hb/HbF, blood transfusions, leukocytosis, and consultation or hospitalization rates significantly correlated with VOC. Three pain-related gene variants correlated with VOC (*CACNA2D3*-rs6777055, *p* = 0.025; *DRD2*-rs4274224, *p* = 0.037; *KCNS1*-rs734784, *p* = 0.01) [30]. Five pain-related gene variants correlated with hospitalization/consultation rates. (*COMT*-rs6269, *p* = 0.027; *FAAH*-rs4141964, *p* = 0.003; *OPRM1*-rs1799971, *p* = 0.031; *ADRB2*-rs1042713; *p* < 0.001; *UGT2B7*-rs7438135, *p* = 0.037) [30]. HbF-promoting-locus variants correlated with decreased hospitalization (*BCL11A*-rs4671393, *p* = 0.026; *HBS1L-MYB*-rs28384513, *p* = 0.01) [30]. *APOL1* G1/G2 correlated with increased hospitalization (*p* = 0.048) [30]. The -α3.7k α-thalassemia deletion correlated with increased VOC (*p* = 0.002) [30], a finding corroborated by a French pediatric SCA study (*p* < 0.001) [26]. Likewise, an independent study showed a significant role of an additional dopamine receptor, *DRD3*, in VOC frequency [60].

A study of *n* = 242 adult HbSS patients (from King’s College Hospital, 33.05 ± 11.26 years) and *n* = 977 children with HbSS or HbSβ^0^ thalassemia (from the SIT trial, 8.98 ± 2.43 years) was conducted to study *PKLR* association with SCD-related pain [61]. Within the adult cohort, 7 *PKLR* SNVs were associated with increased hospitalization rates for acute pain (intron 4, rs2071053; intron 2, rs8177970, rs116244351, rs114455416, rs12741350, rs3020781, and rs8177964), and remained significant after multiple-testing correction (*p* < 0.0071) [61]. Allele-specific expression analysis on a separate cohort of *n* = 59 SCD patients suggested that intronic variants may influence acute pain episodes by affecting *PKLR* expression; however, the mechanism is not yet defined.

For a cohort of *n* = 82 Egyptian SCD patients (53 SS, 29 S/β thalassemia, 12.8 ± 8.4 years) and *n* = 70 healthy controls (20.06 ± 12.15 years), all with genotypes for thrombophilic variants, *MTHFR* C677T was found in 40.2% of the patients and *FVL* G1691A in 13.4%, while 8.5% (*n* = 7) had both [62]. The frequency of acute painful crises was associated with the heterozygosity for *MTHFR* C677T (rs1801133) and *FVL* G1691A variants (*p* = 0.003), suggesting its impact as a risk factor for VOCs [62], a finding in line with increased risk for AVN for rs1801133 in a US study with 107 SCA patients (*p* = 0.00367) [35], for combined consideration of AVN together with stroke and retinopathy in a small Brazilian study with 53 mixed SCA and HbSC patients (*p* = 0.047) [43], and for a study considering the *MTHFR* C677T and the *FVL* G1691A variants in an Indian population of SCD patients (*n* = 150; 16.3 ± 6.5) [63] and controls (*n* = 150; 17.6 ± 6.8). In the latter study, polymorphisms of both thrombotic variants (*MTHFR* C677T and *FVL* G1691A) were associated with higher levels of prothrombin fragment (F1 + 2), D-Dimer, thrombin-antithrombin (TAT), lower level of protein C, and higher pain incidence along with earlier onset age of clinical manifestations and frequency of blood transfusion, compared to SCD patients with normal gene variants [63].

A study of whole-blood gene expression profiles in Bahraini SCD patients with VOC (*n* = 10; 34.9 ± 9.3 years), steady-state patients (*n* = 10; 33 ± 10.82 years), including *n* = 8 healthy controls, identified differentially expressed genes [64]. Of those, *PLSCR4* was further validated to be differentially expressed (*p* < 0.01) with high fold change in the comparison of VOC to steady-state patients. PLSCR4 is known to be involved in erythrocyte membrane deformity, in addition to a potential role as a VOC biomarker according to these results, which, however, still require large-scale validation [64].

A study was designed to evaluate the prevalence of *GSTM1*, *GSTT1,* and *GSTP1* gene polymorphisms among homozygous SCA Egyptian pediatric SCD patients (*n* = 50; 10.1 ± 4.7 years) and *n* = 50 healthy controls free of hemoglobinopathies and their association with SCD severity and complications [65]. The results showed a significant association between *GSTM1* and increased risk of severe VOC (*p* = 0.005) [65]. However, there was no significant association between GST genotypes and SCD-related pain frequency, transfusion frequency, or HU treatment [65].

A case-control study of VOC Bahraini patients (*n* = 127; 15.0 ± 8.4 years) and steady-state patients (*n* = 130; 16.0 ± 9.2 years) performed HPA genotyping to associate the polymorphic variants to VOC phenotype [66]. A multivariate model showed an independent association of *HPA-3a*/*3b* and *HPA-3b*/*3b* genotypes with VOC [66]. *HPA-3a*/*3b* (*p* = 0.002) and *3b*/*3b* (*p* = 0.006) genotypes were associated with the need for hospitalization, while *HPA-3b*/*3b* was associated with VOC frequency (*p* = 0.005), type (localized vs. generalized, *p* = 0.004), and medication (opioids vs. NSAIDs, *p* = 0.002) [66]. Therefore, *HPA-3* appears to be an independent genetic risk factor for SCA VOC. Similarly, a Brazilian case-control study investigated the association of *HPA-1*, *-2*, and *-5* variants for association with wider occlusive vascular complications based on 97 SCA patients divided into closely age- and sex-matched VOC^+^ (*n* = 34) and VOC^−^ (*n* = 63) groups [67]. While no association for *HPA-1* and *-2* genotypes was detected, the presence of *HPA-5b* is strongly associated with VOCs (*p* = 0.0002).

The association of protein Z promoter and intronic SNVs was studied in SCD patients with VOC (*n* = 239; 11.9 ± 7.1 years) and pain-free SCD controls (*n* = 138; 14.2 ± 10.5 years), all Bahraini Arabs, showing that specific PZ variants and haplotypes are associated with SCD-related VOC [68]. Patients with VOCs had a higher frequency of rs3024718 (*p* = 0.03), rs3024719 (*p* = 0.02), rs3024731 (*p* < 0.001), and rs3024735 (*p* < 0.001) variants compared to the steady-state group [68]. Additionally, the distribution of rs3024731 (*p* = 0.028) and rs3024735 (*p* = 0.045) was different between VOC and steady-state patients, even after adjusting for confounders [68]. Finally, the increased frequency of GGTG (*p* = 0.018) and reduced frequency of the AGTG haplotype (*p* = 0.001) in VOC, compared to the steady-state group, remained significant after applying multiple-testing correction [68].

A case-control study by Yousry et al. of *n* = 100 Egyptian SCD vs. 80 control subjects measured VOC frequency over a 2.5-year follow-up period for correlation with two major nitric oxidase 3 (*NOS3*, aka endothelial NOS, constitutive NOS or endothelial constitutive NOS, i.e., *eNOS*, *cNOS* or *ecNOS*) polymorphisms, i.e., -786T > C [rs2070744] and the *NOS3*-*4a/b* 27-bp repeat variation [69]. Homozygosity of the *NOS3*-*4b* allele was associated with the absence of VOCs, whereas, conversely, the presence of the -786C/*4a* combined haplotype was associated with elevated VOCs. In two independent studies, SNPs in GTP cyclohydroxylase 1 (GCH1), which is involved in precursor production for dopamine and serotonin but also in enhancing NO synthesis, and which itself is induced by NO, were also found to be associated with elevated VOC rates [70,71].

### 3.4. Genetic Modifiers Associated with Acute Chest Syndrome

The Egyptian case-control study by Yousry et al., in addition to VOC and other parameters also investigated ACS frequency for the same *NOS3* -786T>C and *NOS3-4a*/*b* polymorphisms [69] to indicate elevated ACS for homozygosity of the -786C allele, as well as for the presence of the -786C/*4a* combined haplotype. A case-control study of African American SCD patients with ACS (*n* = 86; 12.6 ± 4.64 years) and controls (*n* = 48; 14 ± 8.9 years) showed *NOS1* ATT repeat polymorphism association with ACS risk (*p* = 0.001) in patients without physician-diagnosed asthma, and no association with *NOS3* T-786C (rs2070744) [72].

A study of children with SCD (*n* = 942; 5–14 years), recruited from the SIT trial (enrolled across North America and Europe), examined a (GT) dinucleotide repeat located in the promoter region of the *HMOX1* gene for the hypothesis that short alleles are linked to decreased risk of adverse outcomes [73]. Children with 2 shorter alleles (4%; ≤25 repeats) had lower ACS hospitalization rates after adjusting for confounding factors [73]. However, there was no association between allele length and pain rate [73].

*EDN1* (aka *ET-1*) and *NOS3* gene polymorphisms (G5665T and T8002C; VNTR and T-786C) were associated with the occurrence of ACS (in ACS^+^ and ACS^−^ group comparison) and painful VOC (in VOC^+^ and VOC^−^ group comparison) in *n* = 173 children with SCA at the sickle cell center of Guadeloupe [74] and, likewise, for the VNTR polymorphism for the aa/ab genotype (*n* = 28) vs. bb genotype (*n* = 23) in pediatric SCD patients in Egypt [75]. The Guadeloupe study showed that the *EDN1* T8002 allele is associated with increased (ACS^+^: *n* = 95; 11.2 ± 4.2 years vs. ACS^−^: *n* = 62; 10.6 ± 4.8 years; *p* = 0.039), whereas the *NOS3* C-786 allele is associated with decreased risk of ACS (ACS^+^: *n* = 104; 11.2 ± 3.1 years vs. ACS^−^: *n* = 55; 9.9 ± 4.8 years; *p* = 0.021) [74].

A total of *n* = 1514 African American participants from the CSSCD study (14.2 ± 11.9 years) were used to identify new genetic modifiers of SCD severity in relation to ACS and painful VOC [76]. A total of 36 SNVs were identified and genotyped for validation in independent patients from the CSSCD (*n* = 387; 11.2 ± 12.5), SCD patients recruited at Georgia Health Sciences University (*n* = 318; no age information), and patients from the Duke SCD cohort (*n* = 449; 33.7 ± 12.1 years) [76]. An association between ACS and the SNV rs6141803 was identified (*p* < 0.0001) in *COMMD7*, a gene highly expressed in the lung and interacting with nuclear factor-κB signaling to control inflammatory responses [76].

A retrospective cohort study of 250 Brazilian pediatric SCA patients (125 male, 125 female) analyzed SNVs in different genes and regions (*BCL11A*, *HIMP-2A*, and *HIMP-2B HBS1L-MYB* intergenic polymorphisms on chromosome 6, *HBBP1*, and *OR51B5/6*) for a range of SCD-relevant parameters. The study identified a significant association of *HMIP-2*-linked SNVs rs9399137 and rs9402686 with ACS and rs7776196 with pain-related hospitalization [77].

An additional cohort study based on the CSSCD cohort of African American SCD patients investigated the association of polygenic trait scores (PTS) with VOC and ACS rates, as well as other hematological and disease-relevant parameters. An HbF-related PTS combining six SNVs at three loci (*BCL11A* [rs1427407, rs7606173], *HBSL1-MYB* [rs6940878, rs9389269, rs114398597], and *HBBP1* [rs10128556]), analyzed in 1271 genotyped patients, was associated with elevated ACS risk (*p* = 0.0005), but not with elevated risk for VOCs (*p* = 0.21) [78].

A retrospective study was designed to evaluate *NOS3* gene polymorphisms (E298D and T-786C) in *n* = 87 African American SCD patients with (*n* = 41; 36 years) and without (*n* = 46; 31 years) ACS [79]. The D298 allele showed no association, while the minor C-786 allele was significantly associated (*p* = 0.0061) with ACS in females, also when accounting for age as a covariate (*p* = 0.0076) [79]. Therefore, *NOS3* T-786C is a gender-specific modifier associated with ACS relative risk.

A retrospective case-control study with 351 pediatric SCA patients showed the association of ACS occurrence with the frequency of the minor *VEGF* -583T/C (rs3025020) T allele (*p* = 5.5 × 10^−6^ by univariate, *p* = 0.004 by multivariate analysis), which is associated with lower VEGF expression, thus linking the severity of SCA to expression levels of VEGF as a pro-angiogenic factor with a key role in peripheral oxygen supply [80].

### 3.5. Genetic Modifiers Associated with Pain in Priapism

A Brazilian study of SCA and HbSC individuals with (*n* = 37; 12.7 ± 5.5 years) and without (*n* = 51; 14.6 ± 3.5 years) priapism tried to elucidate the mechanism involved in priapism events. It showed that the frequency of *NOS3* and *EDN1* polymorphisms in both groups was not significant (*p* = 0.6 and *p* = 1, respectively) and that the presence of the variant allele was not associated with alterations in NO metabolite and *EDN1* levels [81]. However, multivariate analysis showed that low HbF and NO metabolite were independently associated with priapism (Model 1 *p* = 0.015 and Model 2 *p* < 0.0001) [81]. The lack of statistical significance between the polymorphisms and history of priapism in SCD patients might have been due to the small cohort size, indicating that bigger studies are needed. An additional study for an Indian cohort of 190 male SCD patients of mixed SCA and HbSβ^0^-thalassemia genotype was better powered but did not give an age range and applied an FDR α = 0.1 for multiple testing correction to determine significance across 297 SNVs in 49 genes [82]. For what is an unusually lenient threshold, the study detected statistical significance in three SNVs for association with elevated priapism, *TGFBR3* rs7526590(T), *AQP1* rs10244884(C), and *ITGAV* rs3768780(G) (all with corrected *p* = 0.0896), with roles in inflammation/development, hydration, and cell adhesion, respectively. In a case-control study based on the CSSCD cohort, a subset of male and genotyped SCA patients (selecting *n* = 677 from among 768 eligible) was tested for association of *KLOTHO* (*KL*) SNVs for all patients with priapism (*n* = 148; 25.9 ± 11.6) against significantly younger (*p* < 0.0001) random controls without incidence of priapism (*n* = 529; 21.1 ± 11.6), with significance for *KL* rs2249358 (*p* = 0.0062), *KL* rs211239 (*p* = 0.0072), and the combined *KL* rs211234/rs211239 haplotype (*p* = 7.61 × 10^−5^) [29].

### 3.6. Genetic Modifiers Associated with Avascular Necrosis

A US study based on the SCD cohort from the CSSCD compared osteonecrosis-negative to radiologically confirmed AVN (reported as osteonecrosis)-positive patients, investigating the association of modifiers with AVN in SCD. A shortlist of 7 genes (*BMP6*, *TGFBR2*, *TGFBR3*, *EDN1*, *ERG*, *KL/KLOTHO*, *ECE1*) plus two KL-flanking genes, *STARD13* and *PDS5B* (aka *APRIN*), was investigated, resulting from an initial low-density screen of a list of 66 candidate genes with a role in vascularization, endothelial cells, inflammation, and oxidative stress. Of the equally spaced SNVs across each gene, multiple SNVs for all nine genes showed a statistically significant association of specific genotypes with AVN, establishing a firm role of these genetic modifiers in AVN development [83]. For *BMP6*, this was confirmed by a UK case-control study with mixed SCD genotypes, which detected the association of the minor A allele of the *BMP6* rs3812163 SNV (alone among several *BMP6*, *KL,* and *ANXA2* SNVs investigated) as associated with AVN (*p* = 0.021) [84]. In another look at AVN-linked variants [85], a study of *n* = 1162 patients from two cross-sectional cohorts and *n* = 303 patients from two expression profiling cohorts was conducted [86,87,88,89,90]. Allelic heterogeneity in expression QTL was found for *S100B*, encoding a damage-associated molecular pattern molecule [85], for which the A allele of rs2154586 was associated with AVN, whereas the A allele of rs2070435 was associated with reduced acute pain episodes [85]. Both rs2154586 and rs2070435 were associated with higher *S100B* gene expression, while serum *S100B* concentrations also correlated independently with AVN and reduced acute pain episodes [85].

### 3.7. Genetic Modifiers Associated with Pain in Other or Mixed SCD Phenotypes

Across pain-related phenotypes, the frequency of hospitalization is a reliable indicator of disease severity, and this parameter was reported in a number of studies (see Table 1) [30,58,61,66]. Importantly, additional pain-related phenomena beyond keywords for this review were identified and analyzed across shortlisted studies, but their detailed analysis is not listed in Table 1 to reduce complexity. Such phenomena include, e.g., abdominal pain more generally [40,63,91,92] and cholelithiasis more specifically [33,38,40,92,93,94,95]. For the latter, closely linked variants have been identified, including *UGT1A1*, *HMOX1,* and APOB [33,96,97,98,99].

Other studies investigated pain phenomena of specific interest for this review but did so for statistical analyses in combinations of phenotypes, which precluded them from inclusion in Table 1. For instance, in a cohort study of mostly Indian SCD patients (*n* = 150) with age- and ethnicity-matched controls, a previously published severity index [100] was used for groupwise comparisons, which conflated the disease parameters of chronic pain, painful crises, aseptic necrosis, and priapism (to account for up to 28% of the total score) with other parameters to show the association of three major *NOS3* polymorphisms with differing disease severity. Other studies for *NOS3* investigated specific phenotypes and were thus included in Table 1, including the observation that in severe forms of SCD, individual mild vs. severe *NOS3* SNVs (*NOS3* 4b/a, 894G>T [rs1799983], -786T>C [rs2070744]) were enriched for the mutant alleles (*NOS3* 4a, 894T, 786C) (*p* < 0.0001) in homo- or compound-heterozygosity, as was a combined haplotype of all three (*p* < 0.0001) [101]. In a mixed Indian cohort of adult and pediatric patients, for a total of *n* = 190 SCA patients (104 with normal kidney function (17.8 ± 9.4 years) and 86 with kidney damage (15.0 ± 9.0 years)), no significant differences in genotype frequencies between patients with or without kidney damage were identified (*p* = 0.107) [102]. Additionally, no association of *IL1RN*, *VNTR,* and HbF in determining kidney damage was found (*p* = 0.824 and *p* = 0.781, respectively) [102]. The contradiction of these results with the association of interleukin levels and SCA pathophysiology elsewhere [103,104] indicates that for aspects of the study, further investigation in a more homogeneous or larger cohort may be required.

Cohort size may also have been limiting for other investigations of GDMs affecting pain phenomena. For instance, a retrospective study of *n* = 54 sickle retinopathy children and *n* = 54 controls examined the retinopathy association with different SCD risk factors and manifestations [105]. The study indicated without statistical significance (*p* = 0.054) that G6PD deficiency was more common among retinopathy patients [105]. Despite the small cohort, the study nevertheless found that retinopathy was associated with pain crisis (*p* = 0.011), male sex (*p* = 0.004), and splenic sequestration (*p* = 0.013) [105]. In an association study of *n* = 118 Indian SCD patients (*n* = 98 HbSS and *n* = 20 HbSβ^0^, 15 (10.75–23.25) years (25th to 75th percentile)), the patients were recruited and followed up (5 (3–10) years), mainly to assess the relationship of *KLF1* polymorphisms with comorbidities [99]. The results showed no association with the *KLF1* gene (6 observed polymorphisms, out of which 3 were novel) or XmnI polymorphism, neither for HbF levels nor for SCD-related phenotypes (e.g., ACS, cholelithiasis, leg ulcer, AVN, hepatomegaly, splenomegaly), laboratory parameters, and number of VOC [99].

### 3.8. Network and Enrichment Analysis

The GDMs of SCD-related VOC and pain phenomena were analyzed using the STRING database- [15] to display GDMs in a gene network (Figure 2), calculate their interconnectedness according to currently available evidence (see Appendix A), and test significant enrichment of known pathways in the list of GDMs (see Appendix A). The highest level of interconnectedness within the network is displayed by *COMT* (15 nodes), *OPRM1* (10 nodes), *ADRB2*, *EDN1*, *NOS3* (8 nodes), *DRD2, HBB*, *NR3C1* (7 nodes), and *HBA2* and MTHFR (6 nodes). From among the different classes of pathways tested for enrichment, sufficiently specific and informative GO Process nodes most highly enriched in the network of GDMs were *Response to oxygen-containing compound* (FDR 4.10 × 10^−13^), *Regulation of blood pressure* (FDR 3.74 × 10^−9^), *Nitric oxide transport* FDR 2.69 × 10^−8^), *Response to stress* (FDR 6.42 × 10^−8^), *Carbon dioxide transport* (FDR 5.74 × 10^−7^), and *Response to hypoxia* (FDR 7.90 × 10^−7^).

**Table 1 ijms-26-04456-t001:** Genetic disease modifiers identified as significantly correlated with SCD-associated pain phenotypes.

GDM	SNV(s) ^1^	Phenotype ^2^					Hospitalization	Effect	*p*-Value	Cohort				Ref.
		VOC	Dactylitis	AVN	ACS	Priapism				Diseased Group Size	Diseased Group Age (Age Range)	Control Group Size	Control Group Age (Age Range)	
*ADRB2*	rs1042713	nt	nt	nt	nt	nt	✓	Unclear if increased or decreased	<0.001	436	16 (5–54)	105	Age-matched	[30]
*ANXA2*	rs7163836, hCV11770326, rs7170178, rs1033028, hCV26910500, hCV1571628	nt	nt	✓	nt	nt	nt	CSSCD cohort analysis of genotype enrichment for AVN^+^ (*n* = 442) vs. AVN^−^ (*n* = 455) patients	<0.001–0.034	442	Not stated	455	>20, 6 years older than diseased group	[83]
*APOL1*	G1/G2	nt	nt	nt	nt	nt	✓	Increased hospitalization	0.048	436	16 (5–54)	105	Age-matched	[30]
*AQP1*	rs10244884	nt	nt	nt	nt	✓	nt	Association with occurrence of priapism for the C allele based on FDR alpha of 0.1	0.08959	190	Not stated	/	/	[82]
*AVPR1A*	rs10877969	✓	nt	nt	nt	nt	nt	Increased acute care, in particular for CT genotype	0.02	107	35.2 ± 12.0	/	/	[106]
*BCL11A*	not specified	✓	nt	nt	nt	nt	nt	Unweighted polygenic score showed high association with VOC	<0.0001	722	>11.5	/	/	[55]
	rs4671393	nt	nt	nt	nt	nt	✓	HbF-promoting variants correlated with decreased hospitalization	0.026	436	16 (5–54)	105	Age-matched	[30]
	rs1427407 rs7557939 rs1186868	✓	✓	nt	ns	nt	nt	HbF-promoting variants correlated with fewer pain (and dactylitis, p not given) episodes with placebo [and HU] treatment	0.0066 [<0.0001] 0.0003 [0.0005] 0.0006 [0.0005]	190 [94 HU, 96 PB]	14 months, 9–18 months	/	Age-matched	[33]
	rs4671393	nt	nt	nt	ns	nt	[✓]	Association with increased infection-related [but not pain-related] hospitalization	0.05	250	8.86 ± 0.19 (5–16)	/	/	[77]
*BMP6*	rs270393, rs267196, rs267201, rs449853, rs1225934	nt	nt	✓	nt	nt	nt	CSSCD cohort analysis of genotype enrichment for AVN^+^ (*n* = 442) vs. AVN^−^ (*n* = 455) patients	0.001–0.012	442	Not stated	455	>20, 6 years older than diseased group	[83]
	rs3812163	nt	nt	✓	nt	nt	nt	Case-control study showing close association of the minor BMP6 allele with AVN	0.021	39	38.82 (19–65)	205	35.9 (15–84)	[84]
*CACNA2D3*	rs6777055	✓	nt	nt	nt	nt	nt	Unclear if increased or decreased	0.025	436	16 (5–54)	105	Age-matched	[30]
*COMMD7*	rs6141803	nt	nt	nt	✓	nt	nt	Increased risk for ACS	< 0.0001	1514	14.2 ± 11.9	/	/	[55]
*COMT*	rs4680	✓	nt	nt	nt	nt	nt	Higher chance of VOC for rare Met allele Higher rate for Met over Val homozygotes	0.020 0.010	113	Overall 34.78 ± 11.35	17	Overall 34.78 ± 11.35	[60]
	not specified	✓	nt	nt	nt	nt	nt	Unweighted polygenic score showed high association with VOC	<0.0001	722	>11.5	/	/	[55]
	rs4633, rs165599 (ATCAA haplotype)	nt	nt	nt	nt	nt	✓	Increased frequency of pain-related ER visits, profoundly in women	0.0004	438	36 ± 12.5	/	/	[58]
	rs6269	nt	nt	nt	nt	nt	✓	Unclear if increased or decreased	0.027	436	16 (5–54)	105	Age-matched	[30]
*DRD2*	rs4274224	✓	nt	nt	nt	nt	nt	Unclear if increased or decreased	0.037	436	16 (5–54)	105	Age-matched	[30]
*DRD3*	rs6280	✓	nt	nt	nt	nt	nt	Absence or fewer VOC for heterozygotes (*n* = 52/*n* = 13) vs. homozygote (*n* = 61/*n* = 4)	0.035	113	Overall 34.78 ± 11.35	17	Overall 34.78 ± 11.35	[60]
*ECE1*	rs212527	nt	nt	✓	nt	nt	nt	CSSCD cohort analysis of genotype enrichment for AVN^+^ (*n* = 442) vs. AVN^−^ (*n* = 455) patients	<0.001	442	Not stated	455	>20, 6 years older than diseased group	[83]
*ERG*	rs979091, rs2836430	nt	nt	✓	nt	nt	nt	CSSCD cohort analysis of genotype enrichment for AVN^+^ (*n* = 442) vs. AVN^−^ (*n* = 455) patients	0.005–0.014	442	Not stated	455	>20, 6 years older than diseased group	[83]
*EDN1*	rs5369, hCV7464888	nt	nt	✓	nt	nt	nt	CSSCD cohort analysis of genotype enrichment for AVN^+^ (*n* = 442) vs. AVN^−^ (*n* = 455) patients	0.001	442	Not stated	455	>20, 6 years older than diseased group	[83]
	T8002	nt	nt	nt	✓	nt	nt	Increased risk of ACS	0.039	95	11.2 ± 4.2	62	10.6 ± 4.8	[74]
*FAAH*	rs4141964	✓	nt	nt	nt	nt	nt	Unweighted polygenic score showed high association with VOC	<0.0001	722	>11.5	/	/	[55]
	rs4141964	nt	nt	nt	nt	nt	✓	Unclear if increased or decreased	0.003	436	16 (5–54)	105	Age-matched	[30]
*G6PD*	various	✓	nt	nt	nt	nt	nt	G6PD deficiency correlated with fewer pain episodes for placebo [but not for HU] treatment	0.0010 [0.26]	190 [94 HU, 96 PB]	14 months, 9–18 months	/	Age-matched	[33]
*GCH1*	rs8007267 [rs2878172]	✓	nt	nt	nt	nt	nt	Association with severe VOC for two cohorts, discovery cohort (155 vs. 73)|CSSCD cohort (313 vs. 200) for rs8007267 at FDR ≤ 0.05; [for rs2878172 at FDR ≤ 0.1]	0.02|0.004 [0.01]	155|313	32.4 ± 10.0|n/a	73|200	34.9 ± 13.6|n/a	[71]
	rs3783641 haplotype rs3783641-rs8007267	✓	nt	nt	nt	nt	nt	rs3783641A associated with higher VOC rate (additive|recessive model) Haplotype rs3783641T-rs8007267C with lower VOC rate	0.024|0.018 0.001	131	34.3 ± 11.8	n/a	n/a	[70]
*GSTM1*	not specified	✓	nt	nt	nt	nt	nt	Increased risk of severe VOC	0.005	50	10.1 ± 4.7	50	Age-matched	[65]
*HBA1/HBA2*	various	✓	nt	nt	nt	nt	nt	For patients under HU treatment, ATT correlated with fewer pain episodes	0.033	190 [94 HU, 96 PB]	14 months, 9–18 months	/	Age-matched	[33]
	-α3.7k (type I) IthaID: 300	✓	nt	nt	nt	nt	nt	Increased frequency of VOC	0.002	436	16 (5–54)	105	age-matched	[30]
	-α3.7k (type I) IthaID: 300	✓	nt	nt	nt	nt	nt	Enriched for VOC (allele frequency 0.34 VOC vs. 0.1–0.28 Others)	<0.001	127	Not stated	187	Not stated	[26]
	various	nt	nt	✓	nt	nt	nt	CSSCD cohort analysis of genotype enrichment for AVN^+^ (*n* = 442) vs. AVN^−^ (*n* = 455) patients	0.03	442	Not stated	455	>20, 6 years older than diseased group	[83]
	-α/-α	nt	nt	nt	✓	nt	nt	UK SCA cohort with significant decrease in occurrence of ACS for -α/-α homozygotes (7 ACS^+^, 37 ACS^−^) compared to αα/αα (30 ACS^+^, 58 ACS^−^)	0.016	44	Not stated	88	Age-matched	[27]
	various	nt	nt	✓	nt	nt	nt	US SCA cohort with significant increase in occurrence of AVN for α-thal (24 AVN^+^, 14 AVN^−^) compared to non-α-thal (20 AVN^+^, 42 AVN^−^)	0.00152	24	Overall 34.8 (15–54)	83	Overall 34.8 (15–54)	[35]
*HBB* haplotype	Combined rs7482144, rs2070972, rs2844105, rs10128556, rs968857, rs10768683	✓	nt	nt	nt	nt	nt	Favorable haplotypes correlated with (unexpectedly) more pain episodes for placebo [and not for HU] treatment	0.0201 [0.35]	190 [94 HU, 96 PB]	14 months, 9–18 months	/	Age-matched	[33]
	Combined rs11036351, rs4320977, rs16912210, rs2855039 rs7482144	nt	nt	nt	✓	nt	nt	Lower ACS rate for H1/H3 and H3/H3 haplotypes vs. H1/H1	0.02	99	8.5 (5–14) (across the study)	410	Age-matched	[50]
	Senegal/Benin, Benin/Benin, CAR/Benin haplotypes	✓	nt	✓	nt	✓	✓	Lower hospitalization, VOC, AVN, and priapism rates for Senegal/Benin haplotype	0.002 0.002 0.00001 0.025	28	Not stated	139	Not stated	[41]
HbF-related PTS SNVs	BCL11A [rs1427407, rs7606173] HBSL1-MYB [ rs6940878, rs9389269, rs114398597] HBBP1 [rs10128556]	ns	nt	✓	nt	nt	nt	CSSCD cohort analysis of HbF-related PTS association with ACS rate	0.0004	1271	Not stated	n/a	n/a	[78]
*HBG2*	rs7482144 (XmnI)	✓	✓	nt	nt	nt	nt	High-HbF SNV correlated with (unexpectedly) more pain episodes for placebo [and not for HU] treatment	0.0047 [0.35]	190 [94 HU, 96 PB]	14 months, 9–18 months	/	Age-matched	[33]
*HPA* genes *ITGA2*	3a/3b & 3b/3b	✓	nt	nt	nt	nt	✓	Both associated with increased need for hospitalization; the latter also associated with increased VOC frequency	0.002 & 0.006; 0.005	127	15.0 ± 8.4	130	16.0 ± 9.2	[66]
	5a/5b & 5b/5b ITGA2	✓	nt	nt	nt	nt	nt	VOC risk greatly increased in the presence of the *HPA-5b* allele	0.0002	34	29.4 (16–48)	63	27.8, 14–51	[67]
*IL1A*	not specified	✓	nt	nt	nt	nt	nt	Unweighted polygenic score showed high association with VOC	<0.0001	722	>11.5	/	/	[55]
*ITGAV*	rs10244884	nt	nt	nt	nt	✓	nt	Association with occurrence of priapism for the G allele based on FDR alpha of 0.1	0.08959	190	Not stated	/	/	[82]
*KCNS1*	rs734784	✓	nt	nt	nt	nt	nt	Unclear if increased or decreased	0.01	436	16 (5–54)	105	Age-matched	[30]
*KCNJ6*	not specified	✓	nt	nt	nt	nt	nt	Unweighted polygenic score showed high association with VOC	<0.0001	722	>11.5	/	/	[55]
*KLOTHO*	rs480780, rs211235, s2149860, s685417, rs516306, rs565587, rs211239, rs211234, rs2238166, s499091, rs576404	nt	nt	✓	nt	nt	nt	CSSCD-based case-control study of genotype enrichment for AVN^+^ (*n* = 442) vs. AVN^−^ (*n* = 455) patients	0.001–0.046	442	Not stated	455	>20, 6 years older than diseased group	[83]
	rs2249358 rs211239 combined rs211234/rs211239 haplotype	nt	nt	nt	nt	✓	nt	CSSCD case-control study of genotype enrichment for priapism^+^ (*n* = 148) vs. priapism^−^ (*n* = 529) patients (*p*-values based on OR and CI [107])	0.0062 0.0072 7.61 × 10^−5^	148	25.9 ± 11.6	529	21.1 ± 11.6	[29]
*MBL2*	rs7096206, rs10031251	✓	nt	nt	nt	nt	nt	Low MBL level association with VOC frequency	0.0229	117	5	/	/	[52]
	rs5030737 (R52C), rs1800450 (G54N), rs1800451) (G57E)	✓	nt	nt	nt	nt	nt	Association of AO + OO haplotype with VOC+ group (*n* = 48, 27 AA) vs. VOC- group (*n* = 39, 31 AA)	0.039	48	2.5 median (0–5) (given for 87 total)	39	2.5 median (0–5) (given for 87 total)	[53]
	XY variant [-221G>C] AO haplotype [rs5030737 (R52C), rs1800450 (G54N), rs1800451) (G57E)]	✓	nt	nt	nt	nt	nt	Association of conflated “intermediate/low” (YA/XA, XA/XA, YA/YO) haplotypes [vs. haplotypes “high” (YA/YA)] with high VOC frequency	0.0188	39	3.41 ± 1.58	48	3.52 ± 1.42	[54]
*MTHFR*	C677T and FVLG1691A variants	✓	nt	nt	nt	nt	nt	Increased frequency of acute painful crises	0.003	82	12.8 ± 8.4	70	20.06 ± 12.15	[62]
	rs1801133 [C677T]	nt	nt	✓	nt	nt	nt	US SCA cohort with significant increase for MTHFR CT&TT (16 AVN^+^, 8 AVN^−^) compared to MTHFR CC (29 AVN^+^, 54 AVN^−^) of occurrence of AVN	0.00367	24	Overall 34.8, (15–54)	83	Overall 34.8 (15–54)	[35]
	rs1801133 [C677T]	nt	nt	✓	nt	nt	nt	Brazilian mixed SCA & HbSC cohort with significant association of MTHFR CT&TT with a combination of AVN, stroke, and retinopathy	0.047	19	Overall 32.8 (13–72)	34	Overall 32.8 (13–72)	[43]
*MYB*	not specified	✓	nt	nt	nt	nt	nt	Unweighted polygenic score showed high association with VOC	<0.0001	722	>11.5	/	/	[55]
	rs28384513	nt	nt	nt	nt	nt	✓	HBS1L-MYB HbF-promoting loci variants correlated with decreased hospitalization	0.01	436	16 (5–54)	105	Age-matched	[30]
	rs9399137 rs28384513	✓	✓	nt	nt	nt	nt	High-HbF SNV correlated with fewer episodes differentially: rs9399137 for pain {and dactylitis} for placebo, and rs28384513 for [HU] treatment	0.0001 {0.0149} [0.22] 0.31 [<0.0001]	190 [94 HU, 96 PB]	14 months, 9–18 months	/	Age-matched	[33]
*NOS1*	AAT repeats	nt	nt	nt	✓	nt	nt	ACS risk in patients without physician-diagnosed asthma	0.001	86	12.6 ± 4.64	48	14 ± 8.9	[72]
*NOS3*	C-786	nt	nt	nt	✓	nt	nt	Decreased risk of ACS	0.021	104	11.2 ± 3.1	55	9.9 ± 4.8	[74]
	C-786	nt	nt	nt	✓	nt	nt	Increased risk of ACS in females	0.0061	41	36	46	31	[79]
	VNTR (intron *4a* vs. *4b* polymorphisms with 4 vs. 5 × 27-bp repeat)	✓	nt	nt	nt	nt	nt	Increased frequency of VOCs for aa/ab (*n* = 28) compared to bb (*n* = 23) genotype within SCD population	0.017	28	11.2 ± 3.9	23	10.6 ± 3.4	[75]
*NR3C1*	not specified	✓	nt	nt	nt	nt	nt	Unweighted polygenic score showed high association with VOC	<0.0001	722	>11.5	/	/	[55]
	rs33389 rs2963155 rs9324918	✓	nt	nt	nt	nt	nt	Higher VOC rate for rs33389T (additive|recessive model) Ditto for rs2963155G (additive|dominant|recessive model) Ditto for rs9324918C (additive model)	0.014|0.011 <0.001|0.021|< 0.001 0.021	136	34.0 ± 11.7	/	/	[56]
*OPRM1*	rs1799971	nt	nt	nt	nt	nt	✓	Unclear if increased or decreased	0.031	436	16 (5–54)	105	Age-matched	[30]
*PDS5B*	hCV3118898, hCV11710292	nt	nt	✓	nt	nt	nt	CSSCD cohort analysis of genotype enrichment for AVN^+^ (*n* = 442) vs. AVN^−^ (*n* = 455) patients	0.001–0.014	442	Not stated	455	>20, 6 years older than diseased group	[83]
*PKLR*	rs2071053, rs8177970, rs116244351, rs114455416, rs12741350, rs3020781, rs8177964	nt	nt	nt	nt	nt	✓	Increased hospitalization	0.0071	242, 977	33.05 ± 11.26, 8.98 ± 2.43	/	/	[55]
*PLSCR4*	not specified	✓	nt	nt	nt	nt	nt	High fold change in the comparison of VOC to steady-state patients	<0.01	10	33 ± 10.82	8	Age-matched	[64]
*PNMT*	rs876493, rs2934965, rs2941523	✓	nt	nt	nt	nt	nt	Lower VOC rate for rs876493A, rs2934965T, rs2941523G	0.012, 0.044, 0.017	131	34.3 ± 11.8	/	/	[108]
*PROZ* promoter	rs3024718, rs3024719, rs3024731, rs3024735; GGTG, AGTG	✓	nt	nt	nt	nt	nt	Higher frequency in VOC compared to steady-state patients	0.028, 0.045; 0.018, 0.001	239	11.9 ± 7.1	138	14.2 ± 10.5	[68]
PTS region *HMIP-2A*	rs7776196, rs9399137, rs35786788, rs796983051, rs79651256	nt	nt	nt	✓	nt	✓	Association of rs7776196 with pain-related hospitalization and of [rs9399137] with ACS	0.04 [0.005]	250	8.86 ± 0.19 (5–16)	/	/	[77]
PTS region *HMIP-2B*	rs9402686, rs4895441, rs9494145	nt	nt	nt	✓	nt	ns	Association of rs9402686 with ACS	0.01	250	8.86 ± 0.19 (5–16)	/	/	[77]
*STARD13*	rs538874, rs475303, rs648464	nt	nt	✓	nt	nt	nt	CSSCD cohort analysis of genotype enrichment for AVN^+^ (*n* = 442) vs. AVN^−^ (*n* = 455) patients	0.001–0.029	442	Not stated	455	>20, 6 years older than diseased group	[83]
*TBC1D1*	not specified	✓	nt	nt	nt	nt	nt	Unweighted polygenic score showed high association with VOC	<0.0001	722	>11.5	/	/	[55]
*TGFBR2*	rs1019856, rs934328	nt	nt	✓	nt	nt	nt	CSSCD cohort analysis of genotype enrichment for AVN^+^ (*n* = 442) vs. AVN^−^ (*n* = 455) patients	<0.001–0.023	442	Not stated	455	>20, 6 years older than diseased group	[83]
*TGFBR3*	rs284157	nt	nt	✓	nt	nt	nt	CSSCD cohort analysis of genotype enrichment for AVN^+^ (*n* = 442) vs. AVN^−^ (*n* = 455) patients	<0.001	442	Not stated	455	>20, 6 years older than diseased group	[83]
	rs7526590	nt	nt	nt	nt	✓	nt	Association with occurrence of priapism for the T allele based on FDR alpha of 0.1	0.08959	190	Not stated	/	/	[82]
*TRPA1*	rs920829 CGAGG haplotype	✓	nt	nt	nt	nt	nt	Lower VOC rate for rs920829 GG genotype Higher VOC count for CGAGG haplotype	0.008 0.009	132	34.2 ± 11.8	n/a	n/a	[109]
*UGT2B7*	rs7438135	nt	nt	nt	nt	nt	✓	Unclear if increased or decreased	0.037	436	16 (5–54)	105	Age-matched	[30]
*VEGFA*	rs3025020	nt	nt	nt	✓	nt	nt	Homozygosity for the minor -583T allele increases risk for ACS according to univariate [and multivariate] analysis	0.013 [0.019]	90	13 (2–16)	261	12, 3–16	[80]
Male/female	n/a	nt	nt	✓	nt	nt	nt	CSSCD cohort analysis of genotype enrichment for AVN^+^ (*n* = 442) vs. AVN^−^ (*n* = 455) patients	0.02	442	Not stated	455	>20, 6 years older than diseased group	[83]

^1^ VNTR—variable number of tandem repeats. ^2^ ACS—acute chest syndrome; ATT—α-thalassemia trait; AVN—avascular necrosis, often specified as osteonecrosis; HU—hydroxyurea; ns—not significant; nt—not tested; PB—placebo treatment; PTS—polygenic trait score; VOC—vaso-occlusive crisis; ✓—significant result detected.

### 3.9. Beyond GWAS and Genetics

This scoping review aimed to chart the present stage of knowledge concerning GDMs affecting VOC and pain-related phenomena in SCD and to give pointers for new studies that would move the field forward. In the process, we confirmed that γ-globin elevation universally mitigates disease severity across β-hemoglobinopathies and that α-globin interactions are more complex, with inconsistent effects on sickle cell disease (SCD) pathology. For instance, while α-globin reduction can reduce ACS and priapism risk, it has variable effects on VOCs and vasculopathy. These apparent inconsistencies underline the many factors that potentially modulate VOC and pain phenotypes in SCD, in line with recent reviews investigating other manifestations of β-hemoglobinopathies in the present special issue of the INHERENT Network. Thus, understanding GDMs affecting hemoglobin expression [23], including α-globin [110] and specifically γ-globin expression [16], is just as fundamental to prognosis and treatment choices for VOC and pain phenotypes as it is for nephropathies [111], stroke [112], and pulmonary hypertension [113] in SCD. Moreover, the commonality of factors concerning endocrinopathies [114] and drug and miRNA treatment efficacy [115] in β-thalassemia and SCD indicates that GDMs so far only associated with β-thalassemia, such as SUPT5H [116], may on closer inspection also have a bearing on SCD phenotypes, including VOC and pain. Such crosstalk between factors affecting both β-hemoglobinopathies is most apparent in findings summarized here for variants on the *HBB* locus itself, where in mixed SCD β-globin genotypes, diverse *HBB* genotypes in combination with β^S^ profoundly affect SCD pain severity. As for SCA, compound heterozygous hemoglobin genotypes, such as HbSC and various HbS/β-thalassemia genotypes, have significant inter-group phenotypic variability, depending on β-globin reduction, γ-globin elevation and the possible contribution of hemoglobin variant proteins to the sickling phenotype, whereas the high intra-group variability emphasizes the importance of GDMs instead. Importantly, and as will be discussed in the following two sections, the identification and interpretation of GDMs must be mindful of still further complexities in phenotype development, which include the consideration of confounding or obscuring effects of environmental factors and the exploration of factors that have been identified by functional or hypothesis-driven studies as exerting an impact on pain phenotypes.

### 3.10. The Influence of Environmental Factors

Many articles refer in an often anecdotal manner to the influence of environmental factors, such as acquired diseases, stress, lifestyle choices, and medication, on pain and pain crises in SCD. Such interactions are important to bear in mind, owing to the possible need to flag them as potential confounding factors for GWAS studies of pain-related phenomena in SCD. However, even for the widespread and closely monitored and reported COVID-19 pandemic, which should allow firmer statistics and findings for SCD than would other environmental insults, observations for the effect of the infection itself or of corresponding treatments on pain-related SCD features have not allowed clear conclusions [117]. VOC-related pain was the most common SCD-related complication and cause of hospitalization, followed by ACS, for SARS-CoV-2 infected SCD patients across waves of SARS-CoV-2 variants [118]. Triggered by infection, SCD-related pre-existing challenges, such as chronic inflammation and necrosis, can lead to extreme symptoms in combination with pronounced pain phenotypes [119]. The influence of genetic modifiers on the severity of the infection then further complicates the identification and quantification of environmental influences on SCD VOC, ACS, and other pain phenotypes. As a case in point, three clinical cases of Brazilian SCD patients under 16 years of age presented with severe complications with COVID-19, including ACS, splenic sequestration, pain crises, and requirement of transfusion, likely brought about by the infection in combination with heterozygosity for missense mutations in *TLR7* and *TIRAP*, respectively, two genes required for effective innate antiviral immunity [120]. Even the most comprehensive GWAS for hemoglobinopathies may fail to account for all phenotypically relevant incidental environmental factors and simultaneously account for genetic modifiers modulating their impact.

More readily considered than such incidental factors, but also far more ubiquitously impacting GWAS analysis results with a chance of leading to faulty conclusions, is the use of symptomatic SCD medication, such as medication that influences the detection of pain phenomena directly or indirectly. In this context, the effect of analgesics would be of immediate relevance for the investigation of pain phenomena and associated GDMs. Striking examples here are several genes and associated SNVs influencing the general efficiency of pharmaceutical pain management, including ABCB1, ABCC3, COMT, CYP2D6, CYP3A4/A5, DRD2, DRD3, KCNJ6, SLC22A1, OPRM1, PNMT, TRPA1, and UGT2B7 [30,60,108,109,121]. The association of such genes and their variants with the frequency of pain crises and dosing and frequency of analgesics in SCD [30,122], while clinically important, may, therefore, not necessarily relate to a causative role in SCD pathology but instead to the action of the associated medication. The same considerations in the investigation of SCD pain phenomena must be given to other medications and the genetic modifiers affecting their action. For instance, the effect of different SNVs on HU response and its effect on hospitalization and other parameters in pediatric patients is still elusive [33], and any analyses of pain phenomena in the presence of HU would have to distinguish factors that influence HU responses and their ameliorating effects from those affecting pain pathophysiology more fundamentally. As for incidental environmental factors (rather than medication), such treatment-related environmental factors may be difficult to account for comprehensively, even in well-designed GWAS analyses.

Finally, genes and variants that directly affect critical SCD-relevant aspects of development and physiology, such as mineral uptake and bone density [123], predisposition for asthma [72,124,125], and vitamin D synthesis, uptake, and signaling [123], would also be predicted to have a causative role in aggravating or ameliorating SCD and SCD-related pain. Many of those genetic factors, however, have as yet not been investigated formally in this context, and their assessment would once again be influenced by environmental factors, such as diet, smoking, or sun exposure, respectively, in the aforementioned examples. Identification of such fundamental modifiers and quantification of their effect on VOC and pain severity by GWAS may, therefore, be suitably supplemented by correlation of gene expression in patients or by rationally designed functional investigations. In addition to a controlled environment, the latter would offer the benefit, for experimentation in mice or cell lines, of isogenic models that would avoid genetic diversity and, thus, the inadvertent influence of undefined genetic modifiers on the outcome. The results of such analyses could then, in turn, inform GWAS design and a data-based and more comprehensive definition of confounding factors.

### 3.11. Areas for Exploration of Genetic Modifiers: Complex Factors, Co-Morbidities, and Linked Phenomena

From among the articles initially shortlisted for this review, several represent studies that did not identify statistically significant findings for genetic variants, precluding their inclusion in Table 1, but that nevertheless allow conclusions about and may guide studies to identify potential additional GDMs of SCD pain phenotypes.

For instance, factors modulating the likelihood of VOCs by affecting sickle cell–endothelial adhesion have been analyzed by a range of approaches. Such studies expand on insights into von Willebrand factor, integrin, and thrombospondin as bridging molecules contributing to vaso-occlusion [126,127,128,129]. Here, reduced thrombospondin interaction with endothelial cells might reduce the risk of sickle-cell adherence and, thus, of VOCs, given that masking of thrombospondin with antibodies reduced endothelial adherence of sickle cells [126]. Along similar lines, the interaction between alpha 4 beta integrin receptor on SCD red blood cells and the vascular-cell adhesion molecule-1 (VCAM-1) expressed on endothelial cells may favor the occurrence of VOC, whereas competing interaction by fibronectin/alpha 4 beta integrin would not. Once again, antibody-based experimental interference established effects on adherence by these components [130], while in further evidence, the possible role of these observations in VOC severity was supported by an independent study that showed significant disease correlation of the levels of various blood adhesion components, such as of VCAM-1, ICAM-1, and E-selectin with cohort mortality, and of P-selectin with other disease parameters [131]. In line with the long-established therapeutic action in SCD of the humanized monoclonal P-selectin antibody crizanlizumab [132] and its ongoing FDA approval despite frequent side effects [133], these findings already indicate a role of blood adhesion components in the context of SCD VOCs. What is more, epinephrine-mediated protein kinase A (PKA) upregulation selectively increased adhesion of SCD but not of normal erythrocytes, possibly owing to PKA-dependent phosphorylation of ICAM-4 [134]. Additionally, ICAM-1 adhesion of HbSS erythrocytes in vitro via fibrinogen as a bridging molecule is correlated with hemolysis, pathological redirection of blood flow (shunts), and low γ-globin expression [135], so that the sum of these data promotes the notion that any of the factors affecting endothelial cell/cell adhesion would warrant closer inspection as therapeutic targets, as VOC-related GDMs, and as potential sites of corresponding, naturally occurring disease-modifying mutations.

Just like low γ-globin levels and elevated endothelial cell adhesion, inflammatory responses and components exert a fundamental effect on pain phenomena in SCD. Though the detection of key inflammatory regulators in this context is not consistent [29,84,102], many reported variants fall into inflammatory pathways, highlighting them as an important context for GDMs of VOC and pain in SCD. Here, such variants variably touched on autoinflammation for the *KIAA1109*-*TENR*-*IL2*-*IL21* gene area [59], angiogenesis as a facilitator of inflammatory responses for VEGF [8,80], and several suppressors of inflammation. The latter included *TGFBR2* and *TGFBR3* detection in the context of AVN [83] and priapism [82], as well as KL [29] and *COMMD7* [76] in two independent studies. Interestingly, both KL and COMMD7 are involved in the termination of NF-κB-mediated proinflammatory responses [136,137], so that their detection goes beyond a confirmation of general inflammatory responses as relevant to pain in SCD, to highlight genes related specifically to NF-κB signaling as potential clinically GDMs of clinical significance.

In another line of investigations, studies focused on a link of leukotrienes to ACS in SCD [138] indicated that leukotrienes experience upregulation in SCD and that, in particular, the leukotriene pathway enzyme phospholipase A2 (PLA2) is greatly upregulated during ACS episodes in HbSS individuals [139]. This upregulation preceding and being predictive of ACS [140,141] indicates that variants modulating the level or functionality of PLA2 are likely candidates as GDMs of SCD-related ACS.

Overall, publications reliably showing a correlation of pain-phenotype-relevant phenomena with the absence or presence of specific endogenous factors may prove valuable for the future design of GWAS, both as potential confounding factors and as potential GDMs. In particular, where GWAS is reliant on massively parallel sequencing approaches, genetic variants affecting factors identified in this manner could be more readily flagged as variants of interest for confirmation or follow-up. However, where some of the more complex relationships of acute pain with other phenomena cannot be defined by GDMs or isolatable factors, awareness of such relationships can nevertheless help prevent erroneous or incomplete interpretation.

One of the most complex relationships of acute pain phenomena is that linking them to chronic pain, which has long been underestimated in its overall impact on SCD patients [142]. Though not a focus of this review, identification of GDMs for SCD chronic pain has been achieved [143] against inherent difficulties in the recognition of chronic pain and in its quantification based on questionnaires and tools allowing the calculation of a composite pain index (CPI) [144,145]. For acute pain phenomena, chronic pain is both a co-morbidity and an underlying condition that increases their frequency and potentially their intensity, e.g., by direct interference with brain and nerve function [146]. Conversely, chronic pain is seen in part as a consequence of acute pain events, which may, over time, lead to central sensitization, i.e., changes in the nervous system that increase sensitivity to pain [147]. This is also reflected at the molecular level in that GDMs of chronic pain overlap with those for acute pain phenotypes, e.g., with variants in *ADRB2* [30,148] and *GCH1* [70,143,148] for both. However, the clinical importance of chronic pain and its relationship with acute pain is, in practice, pitched against the difficulty of designing GDM-focused studies that are sufficiently powered while also analyzing both chronic and specific acute pain phenomena. Illustrating this point, one study for *ADRB2* variants found mixed results for chronic pain (a significant CPI increase for rs1042711C, rs11168070G, rs11959427C, and rs1801704C, but a decrease instead for rs1042713A, rs17778257T, and rs12654778A alleles), while finding no significant association with health care frequency as a proxy measure for acute pain [148]. Finally, and more fundamentally than challenges of study design, chronic pain also influences the mood, perception, and mental health of patients, which, in turn, influences the very questionnaire outcomes used for its assessment. In SCD, chronic pain is accordingly associated with frequent depression and anxiety in adolescents and young adults [149], while conversely higher levels of depression are linked to increased pain, increased interference of pain with normal activities, and increased opioid abuse in adults [150]. The complex connection of chronic pain, mental health, and acute pain phenomena in SCD thus impedes the distinction of cause and effect, the identification of specific GDMs in SCD, and the avoidance of confounding factors in the design of GWAS analyses. Though these limitations cannot readily be addressed by study design, their consideration in the interpretation of future studies will be critical.

### 3.12. Limitations of the Present Article

A key limitation of the present review is that of any review focused on specific disease aspects that are impacted by general disease severity. Corresponding searches are blind to all studies that investigate the impact of GDMs on disease severity more generally, without explicit reference to the pathology aspects and search terms under investigation. For instance, no systematic analysis has been undertaken for different combinations of *HBB* mutations in a comparison of SCA with compound heterozygous conditions to assess specifically the impact on pain aspects of SCD. Moreover, as for the most impactful GDMs of both SCD and β-thalassemia, γ- and α-globin, most studies do not refer to the specific search terms for this review. For a more general understanding of GDMs affecting pain, significant findings concerning globin expression in general [23] and α- and γ-globin expression more specifically [16,111], therefore, also need to be considered. Concerning GDMs affecting VOC in SCD, multiple genetic variants have complex effects on VOC phenotypes, including SNVs in immunity-related genes, directly pain-related genes, genes related to thrombophilia, and transcription factors regulating globin expression. Concerning ACS in SCD, key SNVs affect NO levels, *EDN1*, and γ-globin regulators, including gender-specific effects. Similarly, for priapism in SCD, NO-related genes and *EDN1* affect the risk of occurrence, as do variants affecting inflammation, hydration, and cell/cell adhesion, while other pain phenotypes are influenced by SNVs affecting thrombosis, γ-globin, and bone health. Though comprehensive in its selection for appraisal of all suitable articles matching the search terms, the current coverage of different SCD pain aspects, therefore, still feels incomplete because the available studies on SCD pain aspects are often small and underpowered or, for larger studies, often have their focus on other aspects of SCD. After all, certain disease parameters, such as γ-globin expression levels, are more readily quantified and less heterogeneous than pain phenomena, and therefore, the results are more easily interpreted and published.

Like all reviews, this article is affected by ubiquitous publication bias in the original research studies it covers, resulting in the underreporting of non-significant findings. Conversely, even for sufficiently powered studies, the absence of significant differences for GDMs in groupwise comparisons does not indicate the significance of its identity across groups. For GWAS studies in particular, a universal limitation lies in their functional interpretation, which is always colored by the level of characterization of the affected genes and variants and which, unhelpfully for translational research, creates a bias for follow-up studies toward already well-understood genes and variants. Likewise, any pathway enrichment analyses, such as that illustrated in Figure 2 and detailed in Appendix A, are biased toward better-characterized genes and pathways and blind toward as yet unknown networks.

A more systematic approach is thus needed to apply greater power and reduce bias in the scrutiny of all pain aspects of SCD toward clearer prognosis and genetic counseling for carrier couples of SCD, including disease severity, and to harmonize study design, which would allow the combination and direct comparison of different studies. Large study networks, such as INHERENT, are therefore required for consistent rules across studies to avoid duplication of work and for a truly multi-ethnic approach that will identify GDMs across or in specific communities.

## 4. Conclusions

Given their importance as major morbidity and mortality determinants for SCD, VOC and pain phenomena are surprisingly under-characterized for their genetic determinants and confounding factors of corresponding analyses. Additional identification and characterization of GDMs will critically benefit from systematic analysis of known SCD GDMs for SCA and compound heterozygous SCD genotypes, from larger-scale, multi-ethnic studies, such as that conducted under the INHERENT Network [22], and from comprehensive identification and consideration of environmental and disease-state-dependent genetic elements as confounding factors.

## Figures and Tables

**Figure 1 ijms-26-04456-f001:**
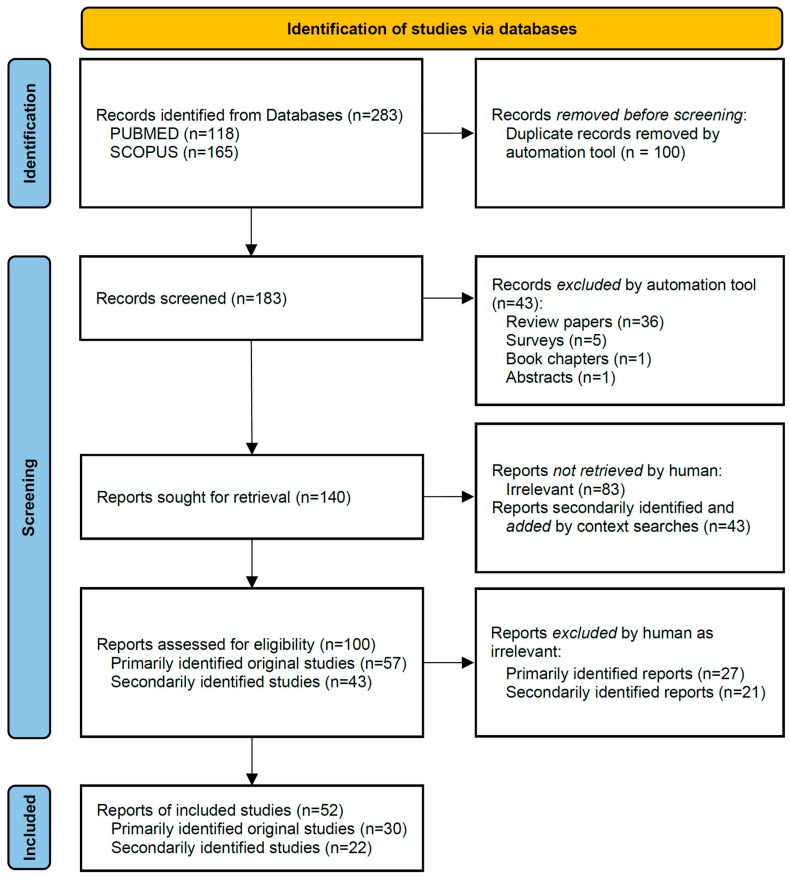
Study selection flow diagram. For Joanna Briggs Institute (JBI) Checklists employed in the selection procedures and assessment of 100 reports for eligibility, see Appendix A. Format based on PRISMA Guidelines 2020 [14], with added inclusion of secondarily identified reports.

**Figure 2 ijms-26-04456-f002:**
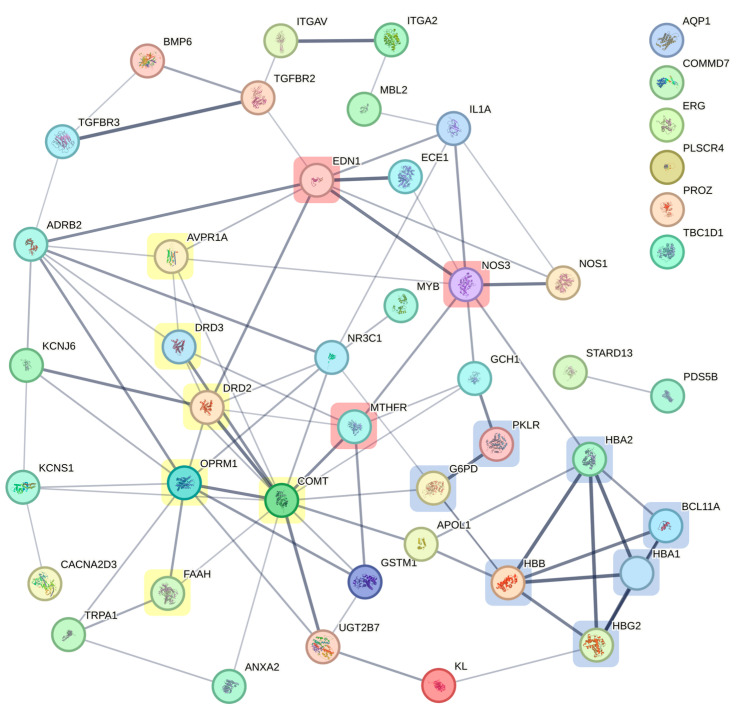
STRING network of genetic modifier genes associated with pain-related phenotypes in SCD. Applying all gene symbols (or their suitable synonyms based on manual correction of the input list), the figure was produced based on the full STRING network, representation of confidence by the network edges, all active interaction sources, medium confidence, and no display of additional interactors. Color overlays as rounded squares indicate exemplary groups of genes involved in DISEASES networks and directly linked by network edges, such as hemolytic anemia (in blue; *p* = 1.36 × 10^−6^), drug dependence (in yellow; *p* = 5.53 × 10^−7^), and hypertension (in red; *p* = 0.0306). Other coloring of symbols is by STRING default. For the statistical strength and the direction of association with VOC and pain phenotypes for specific variants, refer to Table 1 columns ‘*p*-value’ and ‘Effect’, respectively, and to the main text.

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
