# Peer review of "Genetic Modifiers Associated with Vaso-Occlusive Crises and Acute Pain Phenomena in Sickle Cell Disease: A Scoping Review"

_ijms, 2025, doi:10.3390/ijms26094456_

Round 1

Reviewer 1 Report

Comments and Suggestions for Authors

The authors describe the studies that report genetic modifiers that affect VOC in SCD.

The methodology is well described and convincing and the review represents the most up to date knowledge on this specific topic.

Probably, due to length limitations, the authors did not describe or classify the specific way that each gene or gene groups affect or influence in the incidence or severity of VOC (NO, inflammation, coagulability,…) but this classification  is relevant and missing in this article.  I notice that for each gene in the Supplement files the effect is included, but some description in the main manuscript is worthwhile. May be a short table can answer to that.

Author Response

The methodology is well described and convincing and the review represents the most up to date knowledge on this specific topic.

[Response] Thank you for this positive assessment of our article.

Probably, due to length limitations, the authors did not describe or classify the specific way that each gene or gene groups affect or influence in the incidence or severity of VOC (NO, inflammation, coagulability,…) but this classification  is relevant and missing in this article.  I notice that for each gene in the Supplement files the effect is included, but some description in the main manuscript is worthwhile. May be a short table can answer to that.

[Response] Hoping to have understood this comment correctly, we have now added in the “Effect” column of Table 1 the direction of the effect, where this was not already in place and where the information was available in the source article.

Additionally, and to make clear that all significant pathway detections are for enrichment (rather than for depletion) of pathways, we have also added a column “enrichment ratio” in SOM 2, calculated for each #category as follows):

enrichment ratio = (m / 46) / (n / 19,488),

where 46 is the number of input proteins represented in STRING.db, where 19,488 is the number of background proteins for Homo sapiens (https://string-db.org/organism_overview.html)) at the time of analysis, and where n is the observed gene count and m the background gene count for each detected #category. All ratios are above 1, which indicates enrichment; this is also annotated as a remark in SOM 2.

We hope that these two steps address the comment satisfactorily and would otherwise be grateful for further clarification.

Reviewer 2 Report

Comments and Suggestions for Authors

This review article provides information on all the modifying factors of SCD. The authors collaborate with the INHERENT network and have received European funding.
I think it's a great revision job.
I didn't think the authors thoroughly reviewed inflammatory aspects such as the involvement of cytokines or other coagulation factors like fibrinogen and von Willebrand factor. For example, on line 527 the authors only mention interleukin but do not expand on its role.
It might be interesting to add a section on the influence of these factors on SCD as a suggestion.

Table 1: column 1 genes in cursive

Author Response

This review article provides information on all the modifying factors of SCD. The authors collaborate with the INHERENT network and have received European funding.
I think it's a great revision job.

[Response] Thank you for this positive assessment of our article.

I didn't think the authors thoroughly reviewed inflammatory aspects such as the involvement of cytokines or other coagulation factors like fibrinogen and von Willebrand factor. For example, on line 527 the authors only mention interleukin but do not expand on its role.
It might be interesting to add a section on the influence of these factors on SCD as a suggestion.

[Response] While providing more context than is usual for Scoping Reviews, there are still many relevant themes across studies that we did not expand upon. We agree that mention of key coagulation factors and giving focus to inflammation are among the most worthwhile additions. We have now addressed this point by adding general reference to bridging molecules (including fibrinogen, thrombospondin and von Willebrand factor) in the context of cell-cell adhesion and by summarizing key findings in the context of inflammation in a separate paragraph (lines 705 – 717) as follows, hoping that this does the subject area justice.

“Just like low γ-globin levels and elevated endothelial cell adhesion, inflammatory responses and components exert a fundamental effect on pain phenomena in SCD. Though detection of key inflammatory regulators in this context is not consistent [28,83,101], many reported variants fall into inflammatory pathways, highlighting them as an important context for GDMs of VOC and pain in SCD. Here, such variants variably touched on autoinflammation for the KIAA1109-TENR-IL2-IL21 gene area [58], angiogenesis as facilitator of inflammatory responses for VEGF [8,79], and several suppressors of inflammation. The latter included TGFBR2 and TGFBR3 detection in the context of AVN [82] and priapism [81], as well as KL [28] and COMMD7 [75] in two independent studies. Interestingly, both KL and COMMD7 are involved in the termination of NF-κB-mediated proinflammatory responses [138,139], so that there detection goes beyond a confirmation of general inflammatory responses as relevant to pain in SCD, to highlight specifically genes related to NF-κB signalling as a context for GDMs of potential clinical significance.”

Table 1: column 1 genes in cursive

[Response] Done.

Reviewer 3 Report

Comments and Suggestions for Authors

This is an outstanding comprehensive review on this topic-one of the best that I have read. The review methodology is appropriate and well described. The supplemental materials are informative. The results are comprehensive and include a substantial number of non-US studies. The discussion of the influence of environmental factors, co-morbidities, and linked phenomena is novel for such reviews. Limitations and conclusion sections are appropriate. I had a few minor suggestions of additional topics for the authors to consider:

  1. Line 670-The paragraphs describing factors modulating the likelihood of VOCs by affecting sickle cell-endothelial adhesion should also mention P-selectin given the probably therapeutic benefit of Crizanlizumab.
  2. I did not see any discussion of chronic pain, which is common in older teens and adults with SCD, and may complicate the various acute pain phenotypes. Chronic pain might also be considered a co-morbidity that increases acute pain frequency.
  3. Similarly, co-morbid mental health disorders, particularly major depression, impact pain intensity and frequency in SCD, and has identified  genetic contributions, and may also share genetic risks with chronic pain.

Author Response

This is an outstanding comprehensive review on this topic-one of the best that I have read. The review methodology is appropriate and well described. The supplemental materials are informative. The results are comprehensive and include a substantial number of non-US studies. The discussion of the influence of environmental factors, co-morbidities, and linked phenomena is novel for such reviews. Limitations and conclusion sections are appropriate.

[Response] Thank you for this positive assessment of our article.

I had a few minor suggestions of additional topics for the authors to consider:

  1. Line 670-The paragraphs describing factors modulating the likelihood of VOCs by affecting sickle cell-endothelial adhesion should also mention P-selectin given the probably therapeutic benefit of Crizanlizumab.

[Response] We have now added explicit reference to crizanlizumab (lines 692 – 696) and thank the reviewer for helping us prevent what with hindsight would have been a glaring omission.

“In line with the long-established therapeutic action in SCD of the humanized monoclonal P-selectin antibody crizanlizumab [134] and its ongoing FDA approval despite frequent side effects [135], these findings demonstrate the general genetic and therapeutic potential of blood adhesion components in the context of SCD VOCs.”

  1. I did not see any discussion of chronic pain, which is common in older teens and adults with SCD, and may complicate the various acute pain phenotypes. Chronic pain might also be considered a co-morbidity that increases acute pain frequency.

[Response] Prompted by this comment we acknowledge that inclusion of reference to chronic pain phenomena, though outside the main focus of this article, is critical. In consequence, we (a) have added several additional references and table entries reflecting this inclusion and (b) have changed the title to refer to “Acute Pain Phenomena” rather than “Pain Phenomena” as clearer indication to the reader of the main emphasis of the article.

On the topic of chronic pain as a co-morbidity influencing acute pain and interpretation of GWAS results, we have added the following substantial section (lines 733 – 762):

“One of the most complex relationships of acute pain phenomena is that linking them to chronic pain, which has long been underestimated in its overall impact on SCD patients [144]. Though not a focus of this review, identification of GDMs for SCD chronic pain has been achieved [145] against inherent difficulties in the recognition of chronic pain and in its quantification based on questionnaires and tools allowing the calculation of a composite pain index (CPI) [146,147]. For acute pain phenomena, chronic pain is both, a co-morbidity and an underlying condition that increases their frequency and potentially their intensity, e.g. by direct interference with brain and nerve function [148]. Conversely, chronic pain is seen in part as a consequence of acute pain events, which may over time lead to central sensitization, i.e. changes in the nervous system that increase sensitivity to pain [149]. This is also reflected at the molecular level, in that GDMs of chronic pain overlap with those for acute pain phenotypes, e.g. with variants in ADRB2 [29,150] and GCH1 [69,145,150] for both. However, the clinical importance of chronic pain and its relationship with acute pain is in practice pitched against the difficulty of designing GDM-focused studies that are sufficiently powered while also analysing both, chronic and specific acute pain phenomena. Illustrating this point, one study for ADRB2 variants found mixed results for chronic pain (a significant CPI increase for rs1042711C, rs11168070G, rs11959427C, rs1801704C, but a decrease instead for rs1042713A, rs17778257T, rs12654778A alleles), while finding no significant association with health care frequency, as a proxy measure for acute pain [150]. Finally and more fundamentally than challenges of study design, chronic pain also influences mood, perception and mental health of patients, which in turn influences the very questionnaire outcomes used for its assessment. In SCD, chronic pain is accordingly associated with frequent depression and anxiety in adolescents and young adult [151], while conversely higher levels of depression are linked to increased pain, increased interference of pain with normal activities, and increased opioid abuse in adults [152]. The complex connection of chronic pain, mental health and acute pain phenomena in SCD thus impedes the distinction of cause and effect, the identification of specific GDMs in SCD, and the avoidance of confounding factors in the design of GWAS analyses. Though these limitations cannot readily be addressed by study design, their consideration in the interpretation of future studies will be critical.”

  1. Similarly, co-morbid mental health disorders, particularly major depression, impact pain intensity and frequency in SCD, and has identified genetic contributions, and may also share genetic risks with chronic pain.

[Response] Prompted by this comment we had first added a separate section on mental health, to then reduce and merge the content with the section above on chronic pain. We found that overlap with chronic pain phenomena and their assessment made this a logical choice and helped the flow of the text. We hope that this assessment and the content of the addition find approval.